# Combined Effect of Biopolymer and Fiber Inclusions on Unconfined Compressive Strength of Soft Soil

**DOI:** 10.3390/polym14040787

**Published:** 2022-02-17

**Authors:** Chunhui Chen, Kai Wei, Jiayu Gu, Xiaoyang Huang, Xianyao Dai, Qingbing Liu

**Affiliations:** 1Badong National Observation and Research Station of Geohazards (BNORSG), Three Gorges Research Center for Geo-Hazards of Ministry of Education, China University of Geosciences, Wuhan 430074, China; chenchunhui@cug.edu.cn (C.C.); m17762528026@163.com (K.W.); jia_yugu@163.com (J.G.); 2Department of Chemistry, College of Chemistry and Chemical Engineering, Xiamen University, Xiamen 361005, China; hxyisahero@163.com; 3Hubei Provincial Communications Planning and Design Institute, Wuhan 430051, China; hubeidxy@139.com

**Keywords:** biopolymer, fiber, soil, interaction mechanism, combined effect

## Abstract

The utilizing of traditional chemical stabilizers could improve soil engineering properties but also results in brittle behavior and causes environmental problems. This study investigates the feasibility of the combined utilization of an ecofriendly biopolymer and fiber inclusions as an alternative to traditional cement for reinforcing soft soil. A series of unconfined compression tests were conducted to examine the combined effect of the biopolymer and fibers on the stress–strain characteristics, strength improvement, failure pattern, and reinforcement mechanism of soft soil. The results show that the biopolymer associated with fibers has an unconfined compressive strength similar to that of fiber-reinforced soil. However, it then increases with different curing times and conditions, which can be up to 1.5 MPa–2.5 MPa. The combined effect of fibers and the biopolymer is not simply equivalent to the sum of the effects of each individual material. The fiber shows its role instantly after being mixed into soil, whereas the effect of biopolymer gradually appears with sample curing time. The biopolymer plays a dominant role in increasing the peak unconfined compressive strength and brittleness of soil, while the amount of fiber is crucial for reducing soil brittleness and increasing ductility. It is shown that the biopolymer not only contributes to the particle bonding force but also facilitates the reinforcement efficiency of fibers in the soil. The fibers in return assist in reducing the soil brittleness arising from biopolymer cementation and provide residual resistance after post-peak failure.

## 1. Introduction

In geotechnical engineering practice, untreated soft soil with low strength, large deformation or poor stability may lead to slope instability, road pavement deformation and building leaning [1,2]. Soil reinforcement is a technique to improve soil geotechnical properties including shear strength, compressibility, and bearing capacity. The commonly used soil reinforcement techniques are physical and chemical soil stabilization [3,4,5]. Chemical compounds are mixed with soil particles to modify the soil geotechnical properties in the chemical stabilization method. As a traditional, efficient, and inexpensive chemical stabilizer, cement has played an important role in stabilizing soft soil [6]. When cement is mixed with water and other additives in the soil, chemical reactions occur between these compounds [7]. The hydrated cement increases the soil particle interaction and improves the micropore structure, thus changing the soil properties. Cement is competitive in soil stabilization due to its high strength, long-term durability, and low cost. Despite this, cement manufacture has led to a continuous increase in the emission of carbon dioxide, nitrogen oxides, dust, and solid waste, which has caused severe environmental concerns [8,9]. In 2017, China produced 2.4 billion tons of cement, accounting for 58% of world cement consumption [10]. Thus, the search for a sustainable soil treatment is becoming necessary.

To reduce greenhouse gas emissions and solid waste generation, some ecofriendly materials have been proposed as alternatives to conventional cement for soil stabilization [11]. Microbial secretion biopolymers are ecofriendly materials with good thickening properties and excellent stability in acidic, alkaline, and saline environments [12]. The application of biopolymers to soil reinforcement/stabilization has emerged recently. Extensive studies have been conducted on the behavior of biopolymer-reinforced soils, including the swelling index, permeability, compatibility, shear strength, compressive strength, and long-term durability. With increasing biopolymer content, the hydraulic conductivity of soil-biopolymer mixes may decrease by five orders of magnitude [13]. Soldo conducted comprehensive research on the effect of biopolymers on soil stabilization with five different types of biopolymers [14]. Unconfined compression, splitting tensile, triaxial, and direct shear tests were carried out with different biopolymer concentrations and curing periods. The results indicate that a higher biopolymer content did not guarantee high soil strength. Biopolymers need more time to react with soil to reach their best performance. In addition, Hataf concluded that the moisture content and curing durations are the two main factors influencing soil strength [15]. Overall, the existence of biopolymers can improve soil geotechnical properties, which shows great potential for soil strength enhancement [16,17], ground improvement [18], and dust and erosion control [19]. However, similar to cement materials, condensed biopolymers lead to brittle soil and dilative performance at failure. Apart from chemical additives, fiber reinforcement has been widely accepted as an effective technique to improve soil engineering properties [20]. It is a green and economical soil improvement method due to its low cost, easy construction, and sustainability [21]. Fiber-reinforced soil behaves as a composite in which fibers provide great tensile resistance. The randomly distributed fibers interact with soil to increase interface resistance and interlock, which improves soil strength [22,23,24,25]. In addition, the high modulus also restrains soil deformation. When the fiber-soil matrix is subject to external forces, fiber inclusions directly link soil aggregates together. After reaching the peak strength, these fibers can postpone the expansion of cracks [26,27]. Thus, soil residual strength and ductility increase.

Thus, this study tries to add fiber inclusions into a biopolymer soil mixture to improve its strength and brittleness. It can be envisaged that the mechanical properties of the soil reinforced with the fiber and biopolymer mixture are more complex than those reinforced by the individual materials. To this end, a series of unconfined compression tests were conducted on soft soil mixed with the biopolymer and fibers. Mixtures were cured under two different conditions for 0 d, 7 d, and 28 d. The microstructure and failure pattern of reinforced soil samples were analyzed to investigate the interaction mechanisms between the biopolymer, fibers, and soil particles.

## 2. Materials and Methods

### 2.1. Materials

The employed soft soil is a silty clay collected in the vicinity of a road construction site along the Yangtze River in Wuhan, China. The basic properties of the soil are shown in Table 1 and Figure 1. Xanthan gum produced by the Xanthomonas campestris bacterium is used in this study. It is a polysaccharide and soluble in water, which shows a promising trend in improving soil geotechnical properties [28,29,30]. The applied fibers are polypropylene fibers. Each fiber is 20 μm in diameter and 12 mm in length. Its elasticity modulus is more than 4000 MPa, while the tensile strength and elongation are 450 MPa and 20%, respectively.

### 2.2. Sample Preparation and Unconfined Compressive Test

Different amounts of biopolymer and fiber were added to the soil to form a reinforced soil mixture. Samples to be tested can be categorized into the following four groups in Table 2. A schematic diagram of sample preparation is presented in Figure 2. The biopolymer was first mixed with deionized water, and the amount of water used was based on the natural moisture content of the tested soil. A biopolymer gel can be obtained by proper mixing through a magnetic stirring apparatus. The fibers, biopolymer, and soil were then mixed in a blender. These mixtures were carefully transferred to a cylinder with a height of 80 mm and a diameter of 40 mm. Considering that the curing conditions affect the biopolymer performance, mixtures were cured within a sealing bag or exposed to air for 0, 7, and 28 days [13,31]. Finally, unconfined compressive tests were conducted at a 1% strain per minute speed.

### 2.3. Scanning Electron Microscope

To explore the interaction between biopolymer, fibers, and soil, the biopolymer–fiber–soil mixture was cut into pieces with dimensions of 10 mm × 10 mm × 10 mm. After freeze-drying, gold was sprayed on the soil mixture surface and then specimens were observed by SEM.

## 3. Results and Discussion

### 3.1. Effect of Fibers and Biopolymer on Soil Strength

The stress–strain curves of individual biopolymer- or fiber-treated soil specimens without being cured are shown in Figure 3A,B. It can be seen that an increase in the biopolymer or fiber content results in higher unconfined compressive strength (UCS), which can be up to 130 kPa for biopolymer-treated soil and 300 kPa for fiber-treated soil; similar observations were also made by Yi [31] and Chen [32]. The stress–strain curves of fiber-biopolymer-reinforced specimens are presented in Figure 3C,D. In general, there is a trend that the combined use of biopolymer and fibers leads to a higher UCS than the individual use of each additive. However, this trend appears non-obvious for the specimens incorporating high contents of fiber. For example, the UCS values of B0.5%F2% and B2%F2% specimens are merely slightly higher than those of B0%F2% specimens. This implies that the strength of the specimens, without being cured, is mainly controlled by the fiber content. When both the biopolymer and fibers are added to the soil, a viscous biopolymer gel occupies the soil pores and cements soil particles to form aggregates while the randomly distributed fibers provide the bond strength, friction, and interlocking force at the interface between the fibers and soil particles/aggregates [33]. However, biopolymer gel within specimen normally requires a certain amount of curing time for sufficiently binding soil particles together; in other words, the contribution of biopolymer to strength improvement cannot be significantly enhanced with the increase of biopolymer content for the specimen that is not cured. This is especially evident in the case of specimens with a high proportion of fibers, where fibers played a prominent role in improving soil strength.

Table 3 lists sample peak compressive strengths under each condition. The stress–strain curves of the reinforced specimens cured under two different circumstances for 28 days are presented in Figure 4 and Figure 5. For all specimens cured within the sealed bag, the overall trend shows that their compressive stress–strain curves are similar to those of uncured specimens. By comparing Figure 3 and Figure 4, it is clear that the specimens incorporating biopolymer show higher strength after being cured in sealed bag (i.e., wet curing conditions) for 28 days, indicating that the extension of curing time is necessary for activating the interaction between biopolymer and soil particles. In addition, the effect of the curing time on the strength improvement becomes more significant with the increase of biopolymer content. For instance, the UCS for the specimen B2%F0.1% is increased by 80 kPa after 28-day curing whereas that for B0.5%F0.1% is increased by only 20 kPa, implying that sufficient curing time allows more biopolymer gel to take part in binding soil particles. The comparison of Figure 3B and Figure 4B indicates that the curing time has no effect on the strength gain for fiber-reinforced specimens; thus, the effect of the curing time arises from the biopolymer. Biopolymers form hydrogels in the soil, and the presence of -COOH and -CH_3_ can provide ionic interactions, cation bridges, and hydrogen bonding that link soil particles [34]. It can be expected that xanthan gum biopolymers can provide stronger hydrogen or electrostatic bonding by curing [35]. With increasing curing time, the crosslinking of the biopolymer gel with the negatively charged clay surface becomes stronger, leading to an increase in the bonding strength between particles [36,37].

Figure 5 shows that the UCS for both the biopolymer-stabilized and fiber-reinforced soil increases considerably when cured in the air. Compared with the fiber-reinforced specimens, biopolymer-stabilized specimens show a better strengthening effect in terms of improving the peak UCS (600 kPa for non-reinforced soil, 1100 kPa for 2% fiber-reinforced soil, and 1800 kPa for 2% biopolymer-stabilized soil). Figure 5A indicates that the biopolymer mainly affects the peak strength value while Figure 5B shows that the fibers exert their main influence on the residual strength. When cured in dry air for 28 days, the water included in viscous biopolymer gel evaporates and biopolymer solid films gradually form. These biopolymer films bond soil particles to fibers firmly, significantly increasing adhesion between these elements [38]. Such an increase in adhesion contributes to the peak strength gain. On the other hand, biopolymer films shrink and become brittle during air-dry curing conditions, and this brittleness behavior becomes more significant as the curing time and biopolymer content increase [39]. It can be seen from Figure 5 that, regardless of biopolymer content, the stress–strain curve exhibits post-peak softening behavior with increasing fiber content. This is because, at the post-peak stage, the relatively large compressive strain enables the full mobilization of tensile resistance of the fibers, which renders the specimens more ductile and hence improves the residual strength of the soil [40,41]. The test results show that the strength of biopolymer-fiber-stabilized soil varies with the content of each reinforcement material, curing time, and curing conditions. To evaluate the relative contributions of biopolymer and fibers to strength gain in the biopolymer–fiber soil mixture, the following formulae were adopted:$$w_{\left(f\right)}=\frac{{\mathrm{UCS}}_{\left(b+f\right)}-{\mathrm{UCS}}_{\left(b\right)}}{{\mathrm{UCS}}_{\left(b\right)}}\;w_{\left(b\right)}=\frac{{\mathrm{UCS}}_{\left(b+f\right)}-{\mathrm{UCS}}_{\left(f\right)}}{{\mathrm{UCS}}_{\left(f\right)}}$$
where w_(b)_ or w_(f)_ is the USC improvement percentage caused by biopolymer or fiber. UCS_(b+f)_ is the peak UCS of biopolymer-fiber soil mixture, while UCS_(b)_ or UCS_(f)_ is the peak UCS of individual biopolymer- or fiber-treated soil at the same content. The calculated w_(b)_ and w_(f)_ values for specimens cured in air-dry conditions are illustrated in Figure 6 and Figure 7.

In Figure 6, each of the different colored columns represents the w_(b)_ values for specimens mixed with a given biopolymer content (0.5%, 1%, and 2%) and varying fiber contents. It can be obtained from these two figures that (1) when fiber content is fixed, the biopolymer strengthening effect increases with higher biopolymer content, and vice versa. When biopolymer or fiber content increases, more interactions can be generated between these components and soil particles, thus directly increasing their working efficiency [42]. In addition, (2) when biopolymer and fiber content are identical, their strengthening percentages are much different. For example, the peak UCS value of the B0.5%F0.5% sample after 7 days of air-dry curing is 1209 kPa, while those of the corresponding biopolymer-treated soil (B0.5%F0%) and fiber-treated soil (B0%F0.5%) are 1058 kPa and 751 kPa, respectively. Consequently, the fiber or biopolymer strengthening percentages are 14.3% and 60.8%, respectively, which indicates a greater contribution of biopolymer to UCS improvement under the air-dry curing condition (3). At each row (from left to right), biopolymer- or fiber-strengthening efficiency decreases as curing time increases. After the first few days of curing, soil particle contact area is reduced significantly and little additional change can be observed [43]. A curing time of 7 days in dry-air conditions is sufficient for the full mobilization of the biopolymer reinforcement process. A further increase of curing time to 28 days facilitates the synergistic effect between the two reinforcement materials. This change in contact area leads to the reduction of the corresponding biopolymer working efficiency.

### 3.2. Failure Pattern

Figure 8 presents the failure characteristics of unreinforced and reinforced soil specimens under different curing conditions. Without being cured, the unreinforced soil (A1) and biopolymer-treated soil (B1) show a similar failure pattern, with a main wide and long crack present in the sample. Considering that their stress–strain curves are also similar, it may be concluded that the biopolymer did not change the soil brittleness when no curing time is provided. After 28 days of curing in the sealed bag (i.e., wet-air circumstance), a certain amount of water persists within the soil; thus, the samples still show a plastic deformation characteristic. Even so, the biopolymer gradually binds soil particles together, reducing the crack size (B2). When adding fibers into the soil (C1~E1), the fibers in the soil serve as a bridge to connect the soil matrix and reduce crack development, regardless of the accompanying biopolymer content. As a result, the wide crack turns into finer fissures. When the samples are cured in air-dry conditions, they become more brittle. This brittle behavior is more pronounced for the specimens with a higher biopolymer content. Figure 8 shows that the brittle failure pattern of D3 (lower-content biopolymer–fiber soil mixture) changes to the ductile pattern of specimen E3 (higher-content biopolymer-fiber soil mixture). This means that the amount of fiber is crucial for reducing brittleness and increasing ductility. In sum, the addition of the biopolymer changes the soil brittleness and strength, while fibers improve the soil ductility.

### 3.3. Combined Stabilized Mechanisms of Fibers and Biopolymer

The UCS results indicate that the combined effect of fibers and the biopolymer is not simply equivalent to the sum of the effects of each individual material [44]. When cured in the wet-air circumstance for a sufficient time period, the biopolymer is gradually distributed uniformly, allowing biopolymer to fully cement soil particles together and thus increasing soil strength. After days of curing in air-dry conditions, with continuous evaporation of water, the biopolymer gel turns to polymer film that can firmly bond to the soil particles and fibers. The dehydration of the biopolymer occurs at the fiber–fiber, fiber–soil, and soil–soil interfaces. Schematic diagrams of the biopolymer–fiber soil interaction are illustrated in Figure 9 and Figure 10. It has been reported that xanthan gum molecules align as threads and textiles to interact with fine particles [45]. For the case of a single fiber, apart from directly linking soil particles and fibers together, the biopolymer attaches to the fiber surface to change the surface morphology, increasing the surface roughness and friction, as indicated in Figure 9. Consequently, the adhesion and friction provided by the fibers and biopolymer together increase the resistance of the soil to external force.

When the fiber content increases, the biopolymer directly binds fibers together, forming a three-dimensional fiber network structure in the soil (Figure 10). The elastic modulus of fibers is much higher than that of soil and the biopolymer, which leads to inconsistent deformation of fibers and soil and relative movement at the fiber–soil interface. Compared with biopolymer-stabilized soil, fibers provide extra tension force to restrain crack development, linking split soil blocks together and preventing brittle failure. In addition, due to the strong adhesive property of the biopolymer, the friction at the fiber- and biopolymer–cemented soil aggregates also increases significantly. Thus, fibers can still provide a large tensile stress to maintain the residual strength of the sample at the post-peak stage. In general, the biopolymer not only contributes to the particle bonding force but also boosts the reinforcement efficiency of fibers. The fibers in return assist in reducing the brittleness of biopolymer–cemented soil and providing residual resistance after the sample fail at the peak strength.

## 4. Conclusions

In this study, unconfined compressive tests were performed to evaluate the combined effect of fibers and a biopolymer on soil strength. The influences of the biopolymer content, fiber content, curing time, and condition were all explored. The following conclusions can be drawn:
There is a trend that the combined use of biopolymer and fibers leads to a higher UCS than the individual use of each additive. However, this trend appears non-obvious for the specimens incorporating high contents of fiber.Specimens incorporating biopolymer show higher strength after being cured in sealed bag (i.e., wet curing condition) for 28 days, indicating that the extension of curing time is necessary for activating the interaction between biopolymer and soil particles.The UCS for samples preserved in the air-dry condition shows different trends. The peak UCS is much higher than that of non-stabilized soil, which can be up to 1114 kPa~2442 kPa. Biopolymer mainly affects the peak strength value while fibers exert their main influence on the residual strength.The combined effect of fibers and the biopolymer is not a simple sum of fiber and biopolymer strengthening effects. The biopolymer not only contributes to the particle bonding force but also boosts the fiber working efficiency in the soil. The fibers in return help to reduce cemented soil brittleness and provide extra resistance after failure. The biopolymer shows a strong effect on soil strength improvement by stronger hydrogen and electrostatic bonding via curing. The fibers can reduce soil brittleness and increase ductility. They link soil blocks and provide internal tensile force to prevent segregation and complete failure of samples. The addition of the biopolymer changes the soil brittleness and strength, while fibers improve the soil ductility.


## Figures and Tables

**Figure 1 polymers-14-00787-f001:**
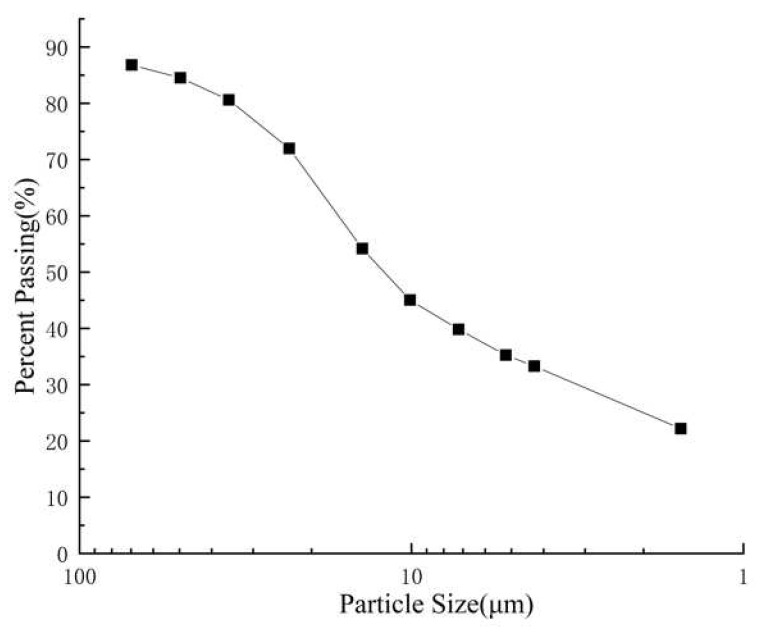
Particle size distribution.

**Figure 2 polymers-14-00787-f002:**
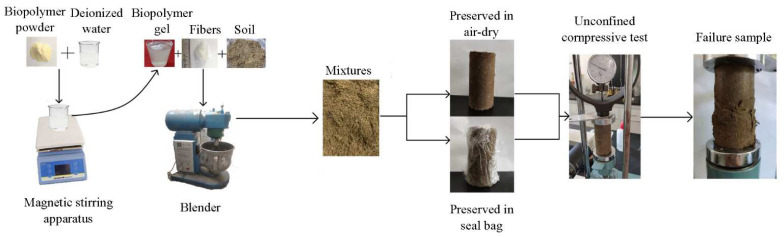
Schematic diagram of sample preparation and test procedure.

**Figure 3 polymers-14-00787-f003:**
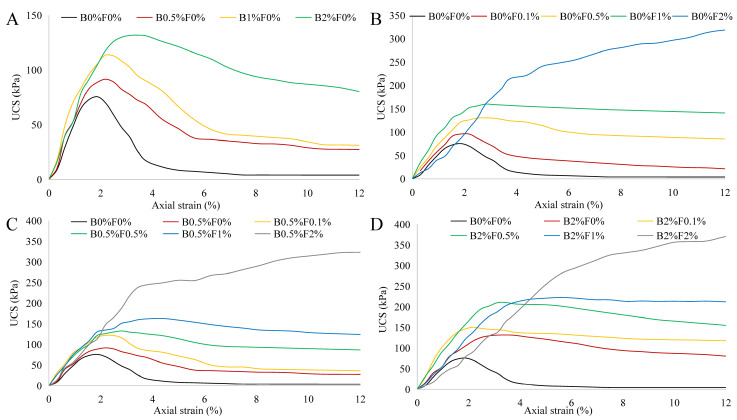
Stress–strain curves of biopolymer- and fiber-treated soil specimens without being cured ((**A**) biopolymer-stabilized soil; (**B**) fiber-reinforced soil; (**C**) biopolymer–fiber-treated soil with low biopolymer content; (**D**) biopolymer–fiber-treated soil with high biopolymer content).

**Figure 4 polymers-14-00787-f004:**
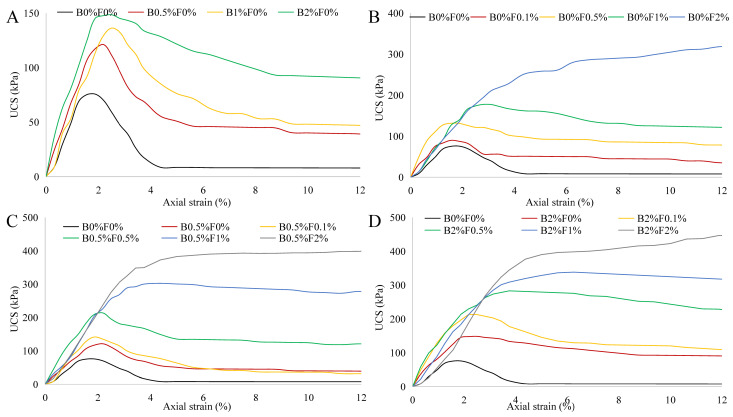
Stress–strain curves of biopolymer- and fiber-treated soil specimens cured in the seal bag for 28 days ((**A**) biopolymer-stabilized soil; (**B**) fiber-reinforced soil; (**C**) biopolymer–fiber-treated soil with low biopolymer content; (**D**) biopolymer–fiber-treated soil with high biopolymer content).

**Figure 5 polymers-14-00787-f005:**
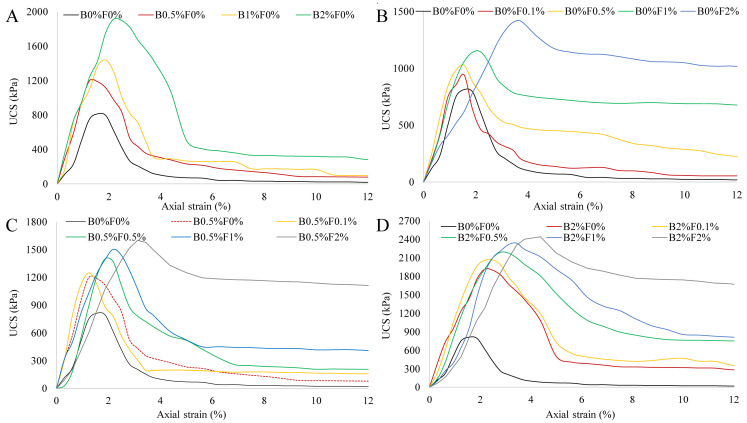
Stress–strain curves of biopolymer- and fiber-treated soil specimens cured in the air for 28 days ((**A**) biopolymer-stabilized soil; (**B**) fiber-reinforced soil; (**C**) biopolymer-fiber-treated soil with low biopolymer content; (**D**) biopolymer-fiber-treated soil with high biopolymer content).

**Figure 6 polymers-14-00787-f006:**
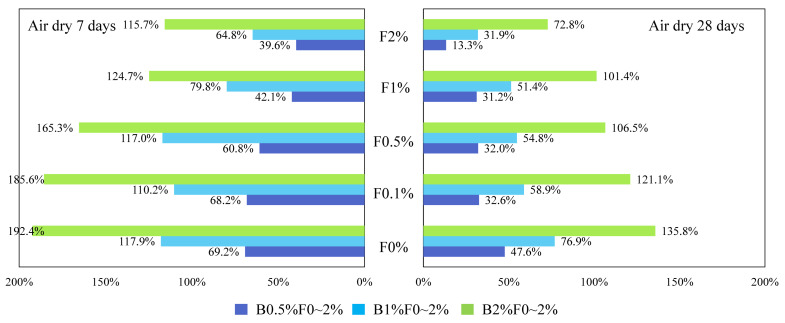
Biopolymer strengthening effect of curing in air-dry-condition specimens.

**Figure 7 polymers-14-00787-f007:**
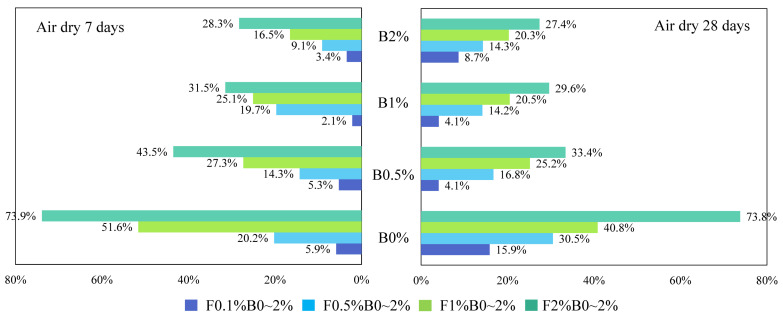
Fiber strengthening effect of curing in air-dry-condition specimens.

**Figure 8 polymers-14-00787-f008:**
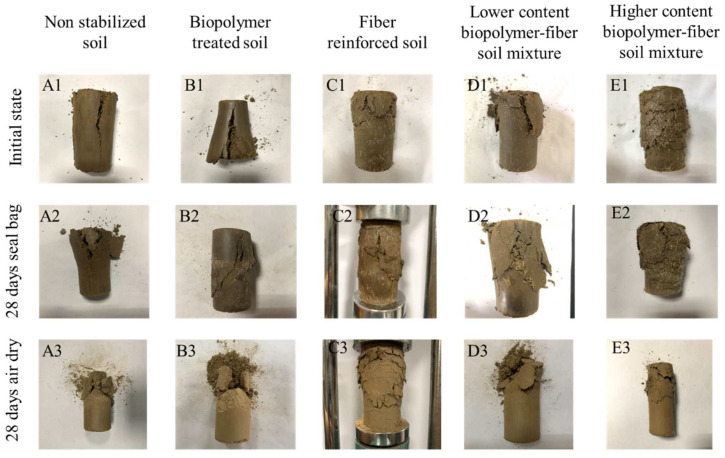
Failure characteristics of non-stabilized and stabilized soil after 28 days curing. ((**A**–**E**) Non stabilized soil, biopolymer treated soil, fiber reinforced soil, lower content biopolymer-fiber soil mixture and higher content biopolymer-fiber soil mixture; (1–3) specimens under initial state, 28 days seal bag curing and 28 days air dry curing).

**Figure 9 polymers-14-00787-f009:**
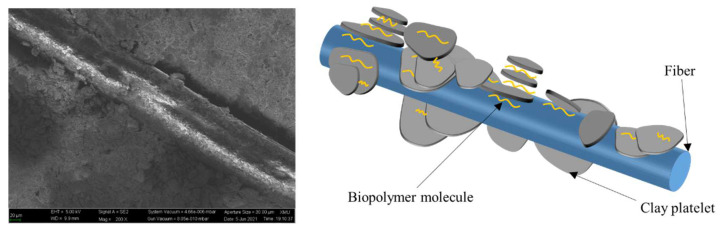
Single-fiber interaction with biopolymer and clay soil.

**Figure 10 polymers-14-00787-f010:**
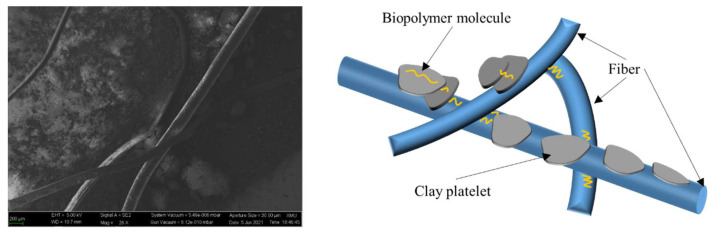
Multiple fibers interaction with biopolymer and clay soil.

**Table 1 polymers-14-00787-t001:** Basic properties of the soft soil.

Indices	Values
Density (g/cm^3^)	2.044
Water content (%)	35.21
Specific gravity	2.75
Plastic limit w_P_ (%)	21.27
Liquid limit, w_L_ (%)	37.13
Plasticity index, PI	15.85
Clay fraction (%)	22.68

**Table 2 polymers-14-00787-t002:** The preparation samples for the test.

Untreated Soil	Biopolymer-Treated Soil	Fiber-Treated Soil	Biopolymer- and Fiber-Treated Soil
B0%F0%	B0.5%F0% B1%F0% B2%F0%	B0%F0.1% B0%F0.5%, B0%F1%, B0%F2%	B0.5%F0.1%, B1%F0.1%, B2%F0.1% B0.5%F0.5%, B1%F0.5%, B2%F0.5%, B0.5%F1%, B1%F1%, B2%F1%, B0.5%F2%, B1%F2%, B2%F2%

PS: B is abbreviation for biopolymer, F is abbreviation for fiber.

**Table 3 polymers-14-00787-t003:** Peak compressive strength for all samples under each condition.

Biopolymer Content	Fiber Content	Initial State	Seal Bag 7 Days	Seal Bag 28 Days	Air-Dry 7 Days	Air-Dry 28 Days	
B0%	F0%	76	79	76	625	813	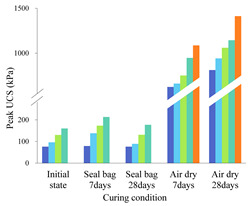
	F0.1%	96	93	89	662	942	
	F0.5%	130	133	131	752	1061	
	F1%	160	163	177	948	1145	
	F2%	/	/	/	1088	1413	
B0.5%	F0%	91	125	122	1058	1200	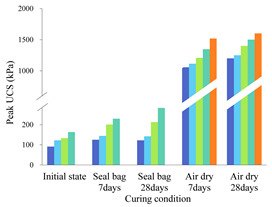
	F0.1%	122	144	142	1114	1249	
	F0.5%	133	201	213	1209	1401	
	F1%	163	230	302	1347	1502	
	F2%	/	/	/	1518	1601	
B1%	F0%	112	136	137	1363	1438	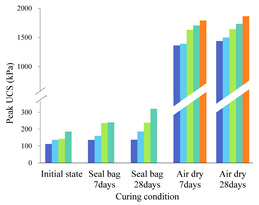
	F0.1%	136	160	186	1392	1497	
	F0.5%	142	236	238	1631	1642	
	F1%	186	240	321	1705	1733	
	F2%	/	/	/	1792	1864	
B2%	F0%	131	152	148	1829	1917	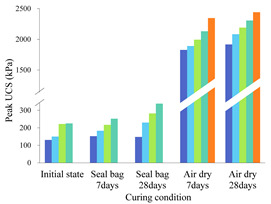
	F0.1%	150	183	230	1891	2083	
	F0.5%	222	217	282	1995	2191	
	F1%	225	252	338	2131	2306	
	F2%	/	/	/	2346	2442	

PS: “/” means the peak UCS cannot be measured because strength keeps increasing.

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
