# Peer review of "Combined Effect of Biopolymer and Fiber Inclusions on Unconfined Compressive Strength of Soft Soil"

_polymers, 2022, doi:10.3390/polym14040787_

Round 1
Reviewer 1 Report
The manuscript is interesting and can be process further; however, I have following Question
- On which basis the different amount of fiber and copolymer were selected.
- It would be more clear if the authors mention the exact amount of used materials, for example water and biopolymer to prepare the gum and so on remaining preparation.
- What is the thickness of the gold layer in SEM?
- It would be helpful to mention all the used abbreviations.
- “Seal bad” is it right? It is table 6 not Figure 6.
- Use same text as used for caption of figures, Fig. or Figure.
- Extend the Figures 7 and 8 length so that text does not overlap.
Author Response
- On which basis the different amount of fiber and copolymer were selected.
Response: The amount of fiber and biopolymer are all commonly used amount in the previous references. Based on these amount of fiber and biopolymer, we are trying to explore their stabilization effect.
- It would be more clear if the authors mention the exact amount of used materials, for example water and biopolymer to prepare the gum and so on remaining preparation.
Response: The revised manuscript now adds a new table (see below) to clearly present the exact amount of the used materials. Please note that the percentage amount shown in the table refers to the ratio of reinforcement material to the dry mass of the soil specimen, and the added amount of water is to ensure that the prepared specimen has a water content of 35.21% (i.e. natural moisture content)
Untreated soil |
Biopolymer treated soil |
Fiber treated soil |
Biopolymer and fiber treated soil |
B0%F0% |
B0.5%F0% B1%F0% B2%F0%
|
B0%F0.1% B0%F0.5%, B0%F1%, B0%F2% |
B0.5%F0.1%, B1%F0.1%, B2%F0.1% B0.5%F0.5%, B1%F0.5%, B2%F0.5%, B0.5%F1%, B1%F1%, B2%F1%, B0.5%F2%, B1%F2%, B2%F2%, |
(B is abbreviation for biopolymer, F is abbreviation for fiber)
- What is the thickness of the gold layer in SEM?
Response: the specimen used for SEM observation was coated with gold layer having a thickness of less than 20nm so as to avoid electron scattering on the specimen surfaces.
- It would be helpful to mention all the used abbreviations.
Response: the abbreviation used in the manuscript is ‘UCS’, which refer to the unconfined compressive strength.
- “Seal bad” is it right? It is table 6 not Figure 6.
Response: We apologize for typos. The ‘seal bad’ has been revised to ‘seal bag’ while the ‘Figure 6’ has been revised to ‘Table 3’.
- Use same text as used for caption of figures, Fig. or Figure.
Response: the revised manuscript now uses the consistent text for the caption of Fig.
- Extend the Figures 7 and 8 length so that text does not overlap.
Response: these two figures have been expanded in the revised manuscript.

Reviewer 2 Report
Thank you for submitting your paper. The work done here draws attention to a significant subject in biopolymer and fiber inclusions in soft soil. I have found the paper to be interesting. However, several issues need to be addressed properly before the paper is being considered for publication. My comments including major and minor concerns are given below:
- Please consider reviewing the abstract and highlight the novelty, major findings, and conclusions. I suggest reorganizing the abstract, highlighting the novelties introduced. The abstract should contain answers to the following questions:
- What problem was studied and why is it important?
- What methods were used?
- What conclusions can be drawn from the results? (Please provide specific results and not generic ones).
- The abstract must be improved. Please use numbers or % terms to clearly shows us the results in your experimental work. Please expand the abstract.
- Please consider reporting on studies related to your work from mdpi journals.
- Please add a list of nomenclature before references for all abbreviations, Greek letters, symbols and letters used in the study. (recommended)
- The introduction can be expanded (it is too short), please consider improving the introduction, provide more in-depth critical review about past studies similar to your work, mention what they did and what were their main findings then highlight how does your current study brings new difference to the field.
- Please enlarge all figures in the manuscript, they are small and somewhat difficult to read. Also, some of them have poor clarity and it is important to improve their resolution.
- Figure 1. Particle size distribution. Change x axis to microns instead of mm for ease of describing the spread.
- 2.3. Fiber combine it with the previous subsection, please avoid adding very small sections of subsections and merge them with the previous or following ones.
- 2.4. Sample preparation and unconfined compressive test some of the data in this subsection can be better presented in a table. For example, the mixing ratios and type of soil specimens used, straining speed and so on. Basically, add all this information as a table of experimental design parameters and their levels.
- Avoid using words such as “obviously” or “clearly” or similar words which do not add any meaningful value to the sentence, please check this everywhere in the article.
- Results where is the discussion section? Or is this meant to be the Results and Discussion section?
- Line 129-130 “concluded that fiber shows more advantageous in strength improvement” this sentence is not clear and requires rephrasing.
- Line 134 “As the fiber content increases, the contribution of biopolymer becomes weaken” this sentence does not read well.
- The paper requires extensive English editing and rephrasing sentences. Many wording does not make sense or is not properly phrased which gives unclear meaning.
- Line 137-138 “reason for this phenomenon is that fibers have a better strengthening effect than biopol..” very poor justification, it does not make sense to say something is better than something, we need to know why in scientific terms, the authors need to provide in depth critical analysis and justification for their findings instead of generic wording which does not add any value to the content of the article or its novelty.
- The authors need to read some article and books about the way fibres behave under loading in order to understand why they are providing an advantage over biopolymer content..etc.
- Line 148 “28d” please use the word days instead of d, check for this elsewhere in the manuscript.
- Line 171 “it presents more brittle behavior with higher initial stiffness” this is not clear, please elaborate further and support with references.
- Line 172 “it decreases markedly at high strain.” what does this sentence mean? The authors need to carefully check their writing style and explanations and they do not make any sense or correctness in terms of English language proficiency.
- Line 174 “The reduction of strength gradually slows down” this very ambiguous statement, what does the authors mean by that?
- Lines 186-208 descriptive paragraphs with minimal scientific justifications or explanations. Please provide in depth discussion rather than just describing the graphs.
- Lines 213-224 combine into one larger paragraph.
- The rest of the article reads the same with exception of 3.4. Combined stabilized mechanisms of fiber and biopolymer and there is minimal scientific discussion or analysis. The authors need to improve the article quality significantly before it can be accepted.
- Some of the results are merely described and is limited to comparing the experimental observation and describing results. The authors are encouraged to include a more detailed results and discussion section and critically discuss the observations from this investigation with existing literature.
- Conclusion can be expanded or perhaps consider using bullet points (1-2 bullet points) from each of the subsections.
Author Response
Thank you for submitting your paper. The work done here draws attention to a significant subject in biopolymer and fiber inclusions in soft soil. I have found the paper to be interesting. However, several issues need to be addressed properly before the paper is being considered for publication. My comments including major and minor concerns are given below:
- Please consider reviewing the abstract and highlight the novelty, major findings, and conclusions. I suggest reorganizing the abstract, highlighting the novelties introduced. The abstract should contain answers to the following questions:
- What problem was studied and why is it important?
- What methods were used?
- What conclusions can be drawn from the results? (Please provide specific results and not generic ones).
- The abstract must be improved. Please use numbers or % terms to clearly show us the results in your experimental work. Please expand the abstract.
Response: Thanks for this insightful suggestion. The abstract is rewritten now.
- Please consider reporting on studies related to your work from mdpi journals.
Response: More papers published in mdpi journals (Applied Sciences/ Materials/ International Journal of Environmental Research and Public Health/ Sustainability) are cited in the revised manuscript now.
- Please add a list of nomenclature before references for all abbreviations, Greek letters, symbols and letters used in the study. (recommended)
Response: the abbreviation used in the manuscript is ‘UCS’, which refer to the unconfined compressive strength.
- The introduction can be expanded (it is too short), please consider improving the introduction, provide more in-depth critical review about past studies similar to your work, mention what they did and what were their main findings then highlight how does your current study brings new difference to the field.
Response: Thank you for this suggestion. The introduction is now rewritten and expanded.
- Please enlarge all figures in the manuscript, they are small and somewhat difficult to read. Also, some of them have poor clarity and it is important to improve their resolution.
Response: Thank you for this suggestion. We redraw all figures and improve their resolution.
- Figure 1. Particle size distribution. Change x axis to microns instead of mm for ease of describing the spread.
Response: The figure is now revised.
- 2.3. Fiber combine it with the previous subsection, please avoid adding very small sections of subsections and merge them with the previous or following ones.
Response: These small sections are now combined.
- 2.4. Sample preparation and unconfined compressive test some of the data in this subsection can be better presented in a table. For example, the mixing ratios and type of soil specimens used, straining speed and so on. Basically, add all this information as a table of experimental design parameters and their levels.
Response: The table is now added and named as table 2
- Avoid using words such as “obviously” or “clearly” or similar words which do not add any meaningful value to the sentence, please check this everywhere in the article.
Response: The mentioned words are now deleted.
- Results where is the discussion section? Or is this meant to be the Results and Discussion section?
Response: It is Results and Discussion section. We revise it now.
- Line 129-130 “concluded that fiber shows more advantageous in strength improvement” this sentence is not clear and requires rephrasing.
- Line 134 “As the fiber content increases, the contribution of biopolymer becomes weaken” this sentence does not read well.
- The paper requires extensive English editing and rephrasing sentences. Many wording does not make sense or is not properly phrased which gives unclear meaning.
- Line 137-138 “reason for this phenomenon is that fibers have a better strengthening effect than biopol..” very poor justification, it does not make sense to say something is better than something, we need to know why in scientific terms, the authors need to provide in depth critical analysis and justification for their findings instead of generic wording which does not add any value to the content of the article or its novelty.
- The authors need to read some article and books about the way fibres behave under loading in order to understand why they are providing an advantage over biopolymer content..etc.
- Line 148 “28d” please use the word days instead of d, check for this elsewhere in the manuscript.
- Line 171 “it presents more brittle behavior with higher initial stiffness” this is not clear, please elaborate further and support with references.
- Line 172 “it decreases markedly at high strain.” what does this sentence mean? The authors need to carefully check their writing style and explanations and they do not make any sense or correctness in terms of English language proficiency.
- Line 174 “The reduction of strength gradually slows down” this very ambiguous statement, what does the authors mean by that?
- Lines 186-208 descriptive paragraphs with minimal scientific justifications or explanations. Please provide in depth discussion rather than just describing the graphs.
- Lines 213-224 combine into one larger paragraph.
- The rest of the article reads the same with exception of 3.4. Combined stabilized mechanisms of fiber and biopolymer and there is minimal scientific discussion or analysis. The authors need to improve the article quality significantly before it can be accepted.
Response: 15-26: Thank you for the suggestion. We find these suggestions are all about section 3.1 and 3.2. We now rewrite these sections and change their structure. More discussions are added to explain the figures in these sections. Besides, the English is edited by AJE English Editing to make it more readable.
- Some of the results are merely described and is limited to comparing the experimental observation and describing results. The authors are encouraged to include a more detailed results and discussion section and critically discuss the observations from this investigation with existing literature.
Response: Thank you for the suggestion. The paper should explain more about results rather than list the data. We add more explanations and references in section 3.1 now. Besides, the section 3.3.may give us some explanations on these results.
- Conclusion can be expanded or perhaps consider using bullet points (1-2 bullet points) from each of the subsections.
Response: Thank you for the suggestion. The conclusions are now revised with bullet points from each of the subsections

Round 2
Reviewer 2 Report
Dear authors,
please improve graphs quality, they are all squashed, you should not copy them as images, they should be added as normal graphs. Looking at them no they are squashed horizontally and do not look good. This is basic article format setup.
Line 19 does not read well at all. The authors write very simple and non sceintific sentences which do not add value to the article: "prevented the biopolymer-treated soil from suddenly failing" this is generic and could mean anything. failing under what load exactly? This is just an example and there are many more in the article. The article needs to be checked thoroughly and carefully for similar sentences and explanations. Make sure all findings and results are explained in a scientific way.
The authors highlighted most of the paper in yellow. Althought much of the text is similar to the original version. For the next revised version, please only show the new changes made in the article and the newly added words/sentences.
Author Response
We replace figures with high resolution and quality, they are not squashed any more. Then, in results and discussion section, the sentences are elaborated in a more scientific way with reference to make it meaningful. We not only list the results but also give further scientific explanations in the test.
Round 3
Reviewer 2 Report
The authors provided the answers to the comments from the second round of review and made sufficient changes in the manuscript according to these comments. I recommend this manuscript for a publication in its present form.
However, the language of the article needs to be considerably improved.